# Collagen Hydrolysate Prepared from Chicken By-Product as a Functional Polymer in Cosmetic Formulation

**DOI:** 10.3390/molecules26072021

**Published:** 2021-04-01

**Authors:** Aneta Prokopová, Jana Pavlačková, Pavel Mokrejš, Robert Gál

**Affiliations:** 1Department of Polymer Engineering, Faculty of Technology, Tomas Bata University in Zlín, Vavrečkova 275, 760 01 Zlín, Czech Republic; mokrejs@utb.cz; 2Department of Lipids, Detergents and Cosmetics Technology, Faculty of Technology, Tomas Bata University in Zlín, Vavrečkova 275, 760 01 Zlín, Czech Republic; pavlackova@utb.cz; 3Department of Food Technology, Faculty of Technology, Tomas Bata University in Zlín, Vavrečkova 275, 760 01 Zlín, Czech Republic; gal@utb.cz

**Keywords:** food by-products, chicken stomachs, collagen hydrolysate, cosmetic gel formulation, aging, hydration, transepidermal water loss, elasticity, roughness

## Abstract

Chicken stomachs can be processed into collagen hydrolysate usable in cosmetic products. The aim of the study was to verify the effects of a carbopol gel formulation enriched with 1.0% (*w*/*w*) chicken hydrolysate on the properties of the skin in the periorbital area after regular application twice a day for eight weeks in volunteers ageed 50 ± 9 years. Skin hydration, transepidermal water loss (TEWL), skin elasticity and skin relief were evaluated. Overall, skin hydration increased by 11.82% and 9.45%, TEWL decreased by 25.70% and 17.80% (always reported for the right and left area). Generally, there was an increase in skin elasticity, a decrease in skin roughness, as the resonance times decreased by 85%. The average reduction of wrinkles was 35.40% on the right and 41.20% on the left. For all results, it can be seen that the longer the cosmetic gel formulation is applied, the better the results. Due to the positive effect on the quality and functionality of the skin, it is possible to apply the cosmetic gel formulation in the periorbital area. The advantage of the product with chicken collagen hydrolysate is also the biocompatibility with the skin and the biodegradability of the formulation.

## 1. Introduction

In 2019, the food industry in the European Union processed more than 15 million tonnes of poultry meat [1]. More than 30% of this is waste material that ends up either in incinerators or in landfills. Secondary solid (bone, chicken stomach, skin) and liquid (blood, fat) raw materials contain proteins (myofibrillar, sarcoplasmic and stromal), lipids (triacylglycerol), vitamins (A, D, E, K, B1 to B12 and vitamin C), carbohydrates (simple and complex) as well as minerals (Ca, K, Na and Mg). The waste contains a large amount of collagen, so the effort is to utilize these products for the production of collagen hydrolysates, which can be used in both the food (due to high nutritional value) and cosmetic industry (anti-aging creams) [2,3]. Collagen hydrolysates are produced from animal tissues that contain significant amounts of collagen in their structure [4]. Molecular weight (M_w_) ranges usually from 0.5 to 15.0 kDa, the hydrolysates are non-toxic and dermatologically acceptable [5]. Further, they are soluble in cold water, have surface-active properties and neutral pH with good biological properties [6]. The chicken stomach is an unpaired organ that is anatomically made up of smooth muscle and has a muscular and glandular part. Chicken stomachs contain a large amount of protein, with a total protein content of 763.96 g/kg in dry matter [7]. With suitable processing of waste into hydrolysates and gelatins, their subsequent application in the food, pharmaceutical and also cosmetic industries is possible. The products contain significant amounts of collagen, elastin, vitamins, enzymes and minerals that the human body needs for life [2].

Natural and synthetic polymers are an important component of cosmetic formulations. Natural polymers include collagen, wheat and soy proteins, hyaluronic acid and others [8]. Cosmetic formulations are further enriched with vitamins, antioxidants, humectants, flavonoids, betacarotenes or retinoids [9,10]. With the help of moisturizers, transepidermal water loss (TEWL) and skin hydration can be improved, thereby reducing the number of wrinkles. These moisturizers are usually applied with creams that contain lipids that increase the barrier function of the skin, water-binding humectants and preservatives. Vitamins A, C, D and E and retinoids (collective name for retinoic acid and vitamin A compounds) have a positive effect on wrinkle reduction. In particular, vitamin A, which reduces fine wrinkles, is used in cosmetic formulations in amounts of up to 1% [10,11]. Another important vitamin is vitamin C (ascorbic acid), as it serves as an essential cofactor for the enzymes lysyl hydroxylase and prolyl hydroxylase, which are required for post-translational processing and biosynthesis of type I and III collagen. Vitamin C increases collagen production and thus reduces the depth of wrinkles and improves the firmness of the skin [11]. The cosmetic gel formulation is a semi-rigid transparent formulation; it contains more water, and as soon as the excess water evaporates after application to the eye area, the gel forms a film that has a cooling effect [12]. Hydrogel are support systems and carriers for active substances [13]. The so-called hydrolysed proteins are mainly used in cosmetic formulations. The most commonly used include peptides, a chemical compound of organic origin that is formed by linking amino acids with a peptide bond [14]. Cosmeceutics can be divided into signal, carrier, peptides as inhibitors of neurotransmitters and peptides that inhibit enzymes [15]. Signal peptides stimulate protein matrix production and collagen synthesis, such as *N*-acetylcarnosine, palmitoyl tripeptide-1 and palmitoyl pentapeptide-4. Carrier peptides supply and stabilize trace elements (copper and manganese) that are essential for wound healing and enzyme synthesis, such as Cu-tripeptide. Peptides as inhibitors of neurotransmitters prevent muscle contraction and thus the formation of wrinkles, such as acetyl hexapeptide-3, pentapeptide-3 and pentapeptide-18. Enzyme-inhibiting peptides act on skin cells through indirect inhibition of enzymes, such as soy and rice peptides [16].

The aging of the organism, especially the skin, is an irreversible phenomenon that is caused by both internal (physiological and genetic predisposition) and external factors (especially the influence of sunlight, but also general lifestyle factors). Due to the action of these factors, negative effects such as itching, dryness, fragility, pigmentation, acne, skin irritation with redness, changes in permeability and increased formation and deepening of wrinkles gradually appear on the skin. With increasing age, mechanical properties decrease, especially the strength and elasticity of the skin, which results in wrinkles and sagging of the skin, which are closely related to the structure and amount of collagen and elastin in the *dermis layer* [8,17]. That is why today’s sophisticated cosmetic products use highly effective active ingredients that significantly improve the appearance and especially the functionality of the skin. As a result of these new product requirements, testing their effectiveness has become an indispensable part of development. Cosmeceutics are defined as products of the cosmetics industry with bioactive ingredients that are expected to improve the appearance of aging skin [18,19]. Cosmeceutics can accelerate the visible results of cosmetic skin care; increased collagen production will relax mimic wrinkles and further improve hydration, barrier function and skin elasticity. The penetration of the active ingredients into the skin is a key factor for the effectiveness of the cosmetic product. Some substances belong primarily to the *stratum cornea* layer (*epidermis*), but others can penetrate to the *dermis* layer. Penetration capacity depends on physicochemical properties, time of action, skin thickness, site, area, skin metabolism, amount of cosmetic formulation and time of application [19]. There are a number of active substances that can effectively affect permeability, the most common of which are carrier peptides and liposomes [20]. Another mechanism for increasing penetration is the use of physical methods such as lasers, iontophoresis, sonophoresis and dermabrasion. These methods alter the structure or electrical conductivity of the skin, thus overcoming the protective functional skin barrier and thus penetrating the cosmetic formulation into the lower layers of the skin [11,21]. Cosmetic formulations based on natural polymers are very advantageous and desirable, as these products are both biocompatible with the skin and biodegradable, as well as environmentally friendly [8,19].

The positive effects of collagen hydrolysates prepared mainly from bovine and porcine sources, but also from fish, on the hydrating, barrier and bio-mechanical properties of the skin were previously demonstrated. In topical applications, low M_w_ hydrolysates (most authors report M_w_ < 5 kDa, some up to 8 kDa) penetrate the skin more easily, accumulate in it, retain water and contribute to the formation of a new collagen biomatrix [22,23]. A significant improvement in the hydration and mechanical properties of the skin after four weeks of topical application of the moisturizing lotion with collagen hydrolysate was recorded [24]. On the contrary, collagen hydrolysates with a higher M_w_ show rather a barrier effect on the skin after topical application of the cosmetic preparation. Such an effect was expressed by the decrease in TEWL due to the formation of a protective film on the skin [25]. The beneficial effects of collagen hydrolysates or peptides on the monitored skin properties were also demonstrated with oral dosing of such products (in the amount of 2.5–5.0 g per day for 2–6 months). Compared to placebo, after regular use of collagen products, higher skin hydration, higher skin elasticity, reduction of wrinkles in the eye area, higher density of the dermal matrix were found [26,27,28]. Some authors also report higher levels of type I procollagen and elastin after eight weeks of taking bioactive collagen peptides [29]. Upon digestion, hydrolyzed collagens are cleaved into oligopeptides or amino acids; these low M_w_ products are then transported by the bloodstream into the body as well as into the dermis. Oligopeptides stimulate the production of new collagen, elastin and hyaluronic acid; amino acids serve as building blocks for the formation of new collagen and elastin [30]. Studies on the positive effect of both oral intake and topical application of collagen peptides on skin biomechanical parameters are also known [31].

## 2. The Objectives and Hypothesis of the Work

Collagen hydrolysates used in cosmetic formulations are mainly prepared from pork and bovine skins and bones. Extractions of hydrolysate from poultry tissues are already known, but no literature has yet mentioned the application of chicken hydrolysates in cosmetics; this is a new unexplored area of use [32,33]. Therefore, the aim of the study was to prepare and test the effects of hydrolysate prepared from chicken muscular stomachs added to a cosmetic gel formulation and to objectively evaluate the effect on the skin using non-invasive bioengineering methods. The monitored skin parameters were skin hydration, TEWL, skin elasticity, change in skin relief or the effects of wrinkle reduction and skin roughness. Scientific hypothesis: Improvement of studied parameters in the periorbital area during eight weeks of regular application twice a day (morning and evening) of cosmetic gel formulation with 1.0% chicken collagen hydrolysate. It is expected to increase skin hydration, reduce TEWL and increase skin elasticity. Another expectation is a decrease in skin roughness and a reduction in the amount of wrinkles in the periorbital area.

## 3. Results and Discussion

### 3.1. Skin Hydration

Figure 1 shows that long-term application improves skin hydration on both right and left temple. Hydration values before application of the cosmetic gel formulation with chicken collagen hydrolysate (week 0) indicate sufficiently moisturizued skin as the avarage values in volunteers were more than 45 c.u. These data indicate that skin hydration values in the periorbital region increased from 54 ± 8 c.u. to 61 ± 19 c.u. on right temple and from 53 ± 12 c.u. to 60 ± 19 c.u. on the left temple. The data also show that there was an overall improvement in the state of skin hydration by 11.82%, respectively by 9.45%. Thus, it has been shown that direct contact of the cosmetic product with the skin surface can increase skin hydration. A very important factor that not only affects skin hydration is a source of collagen. In the case of collagen extracted from a fish source, a deeper penetration into the skin was monitored, which improved the water binding and thus the degree of skin hydration [34]. In a study by Rensburg et al. skin hydration was measured in 21 women with dry skin aged 20 to 60 years in different areas of the face (forehead, chin, face, temples and upper jaw) and in different timelines, short-term (1 h) and long-term (3 weeks). As in the previous study, a positive effect was demonstrated in both short-term and long-term application, and it was also found that in the area of T-zone (forehead and chin) skin hydration reached lower values compared to the face, temples and upper jaw, in which the hydrating effect was much more effective [35]. Mac–Mary et al. conducted an interesting study on forearm hydration in healthy individuals. They found that drinking 1 L of water a day for 42 days increased hydration by 14% [36]. Bolke et al. studied the influence of bovine collagen peptides after 12 weeks of oral intake onto skin parameers. Comparted with placebo drinking 2.5 g of collagen daily resulted in improved skin hydration, elasticity and density; the depth of wrinkles were reduced [27].

### 3.2. Transepidermal Water Loss (TEWL)

Figure 2 shows that long-term application of the cosmetic gel formulation improves the barrier properties of the skin in the periorbital area, thus reducing the permeability of water through the skin, indicating that chicken collagen hydrolysate has a positive effect not only on hydration but also on TEWL. For both hydration and transepidermal water loss, the longer the cosmetic gel formulation is applied, the better the results. Initially, on week 0, TEWL was measured prior to application of the cosmetic gel formulation with chicken collagen hydrolysate. According to the TEWL values, skin condition can be divided into very good (0–10 g/m^2^/h), good (10–15 g/m^2^/h), normal (15–25 g/m^2^/h), tight (25–30 g/m^2^/h) and critical (>30 g/m^2^/h) skin condition. The lower the TEWL, the healthier the skin; TEWL values of 0–10 g/m^2^/h represent very healthy skin and 10–15 g/m^2^/h indicates healthy skin. All participants had normal skin condition because the mean TEWL values were 17.2 ± 0.6 and 16.3 ± 0.6 g/m^2^/h (right and left sleep). In the fourth, but also in the eighth week, there was a desirable decrease in TEWL values, i.e., an improvement in the barrier functions of the skin by an average of 9.8% and 25.7% on the right temple and by an average of 5.0% and 17.8% on the left temple. It is scientifically proven that with longer application times, skin hydration increases and the skin’s barrier function improves; thus, it also increases the water content in the skin and decreases the TEWL, which was also confirmed in this study. In a study by Samad et al. the TEWL values on the neck and forehead were measured. Participants applied the cream for several weeks and then the barrier effect of the cosmetic gel formulation was monitored. During the first three weeks of exposure, the average TEWL values increased fivefold, then kept increasing over three weeks and then remained almost constant. The extreme increase in TEWL values is attributed to the fact that volunteers were in contact with heavy metals and chemicals at work [37].

### 3.3. Skin Elasticity

In Table 1 and Table 2 we can see the average values of resonance times (RRT) with standard deviations in individual angles of rotation of the probe for the right and for the left temple areas, respectively. First, an initial measurement was performed at week 0 and then the measurement was checked after regular application at weeks 4 and 8. The tables show that in addition to the rotation angle of 30° (fourth and eighth week), 60° (fourth week) and 270° (eighth week), there was a decrease of RRT in all angles of rotation at both right and left temple, and thus an overall increase in skin elasticity, as 85% of the RRT values showed an overall improvement. The most effective reduction in elasticity was at a rotation angle of 120° and 150°. At an angle of 120°, the average RRT (right and left area) decreased from 323 ± 60 a.u. to 179 ± 30 a.u. in the fourth week of measurement. At an angle of 150°, the average RRT in the periorbital region decreased from 354 ± 45 a.u. to 196 ± 40 a.u. (fourth week).

In one study by Aguirre–Cruz et al., a cosmetic formulation with different concentrations of collagen hydrolysate (5%, 7% and 10%) was applied to 30 women for 30 days. The results showed that the cosmetic formulation with the highest percentage of collagen hydrolysate (10%) had the best effect not only on skin elasticity but also on skin hydration. Using a larger amount of hydrolysate would probably also improve the effectiveness of the cosmetic formulation, as the study shows that the concentration of collagen hydrolysate is crucial when applying the cosmetic formulation to the skin [38]. The viscoelastic properties of the skin depend on age. Ruvolo et al. focused on typical RRT anisotropy profiles recorded from the upper inner side of the arm for three different age groups. It was found that the RRT anisotropy values increased over time; the changes between age groups were statistically significant [39].

### 3.4. Skin Relief

#### 3.4.1. Roughness Parameters

From the results shown in Figure 3 and Figure 4, it can be seen that the application of the cosmetic gel formulation with chicken collagen hydrolysate reduced wrinkles. The average values calculated from the values of the right and left temple areas are as follows; within the roughness R1 the reduction of wrinkles decreased by 38.0%, the roughness R2 decreased by 42.5%, the roughness R3 decreased by 38.5%, the roughness R4 by 34.0% and the roughness R5 by 38.5%. The average reduction of wrinkles or the average decrease of all roughness parameters was 35.4% on the right side and 41.2% on the left side. In the figures, the roughness decreases according to the assumption R1 ≥ R2 ≥ R3 ≥ R4 ≥ R5 and also within the given week there is a decrease in the studied quantity. The most intense decrease in roughness on both right and left temple is then visible for parameter R1 (distance between highest and lowest value) and R2 (maximum parameter). Parameter R1 decreased on both right and left temple from 0.209 ± 0.023 to 0.130 ± 0.012. Subsequently, R2 decreased on both right and left temple from 0.162 ± 0.015 to 0.131 ± 0.013. The roughness parameter R3 decreased from 0.115 ± 0.010 to 0.071 ± 0.006. For parameter R4, there was an average decrease from 0.087 ± 0.005 to 0.057 ± 0.006 (right and left area). Finally, R5 decreased from 0.032 ± 0.006 to 0.020 ± 0.004.

Cartigliani et al. tested a cosmetic product based on collagen cream against aging in women of different ages (31–70 years) and the roughness parameters R3 and R5 were monitored. It was shown that the values of roughness parameters increased in women with increasing age, and it was also shown that long-term application of the cosmetic formulation in women aged 50–61 years reduced wrinkles as in this study. There was also a reduction in wrinkles in women aged 31–50 and 61–70, but the process was not as intense as in women above the age of 50 [40]. In a study by Ryu et al. the efficacy of the application was observed and subsequently the efficacy of succinyl atelocollagen and adenosine in the periorbital area was compared. The study involved mainly older women and the study lasted two months. They found that succinylated atelocollagen has better solubility compared to adenosine and thus penetrates better into the layers of the *epidermis* and *dermis*. At the same time, succinylated atelocollagen also increases skin hydration, as it retains water more than adenosine. The roughness parameters R3 (Rz) and R5 (Ra) were also studied, and in both cases there was a decrease in the parameters, i.e., an improvement in the relief of the skin as in this study [41].

#### 3.4.2. Evaluation of the Skin Replica Image

In Figure 5 and Figure 6, we can compare images of the skin surface before application (a) and after eight weeks of application (b) of the cosmetic gel formulation to the skin of volunteers aged 45 and 50 years. The pictures show that in both cases there was a reduction in wrinkles, as the depressions were reduced, and thus a smoother surface of the skin was achieved due to the lower intensity of the shadow in the individual images. It follows that long-term application twice a day (morning and evening) of a cosmetic gel formulation with 1.0% chicken collagen hydrolysate will improve the condition of the skin. Measurement of skin microrelief is an interesting indicator for the cosmetics industry, as it is easy to use and economically beneficial for research laboratories and institutions dealing with the effectiveness of anti-aging products.

In 3D images in Figure 7 and Figure 8 we can compare the skin surface of volunteers aged 45 and 50 years before application (a) and after eight weeks of application (b) of cosmetic gel formulation with chicken collagen on the skin. Again, it is clear from the images that in both cases there was a desired reduction in wrinkles (wrinkle smoothing), as the wrinkle depressions were reduced, and thus a smoother skin surface was achieved by long-term application of a cosmetic gel formulation in the periorbital area.

Proksch et al. tested the effect of oral dosing of bioactive collagen hydrolysate on the reduction of temple area wrinkles in volunteers aged 45–65 years. One group took collagen, the other group an ordinary placebo. Wrinkles were reduced by 7.2% after four weeks and by 20.1% after eight weeks compared with the group using placebo. In the presented study (see Section 3.4.1), there was a more significant average overall reduction of wrinkles by 35.4% (right side) and by 41.2% (left side). Thus, oral use of the hydrolysate had a positive effect on wrinkle reduction due to a direct effect on the extracellular matrix, as evidenced by an increase in collagen and elastin synthesis in the skin, but did not have such an effect as the the application of a cosmetic gel formulation [28]. In another study by the same group of authors the effect of oral dosing of specific collagen bioactive agents on the reduction of eye wrinkles was studied. With increasing amounts of bioactive collagen peptide, the amount of wrinkles in the ocular area decreased compared to placebo. Overall, wrinkles were reduced by 17.7% compared to placebo. It was also found that there was an increase in the amount of collagen and elastin in the skin. During the eight-week application, collagen and elastin increased 1.25-fold and 1.20-fold compared to placebo [29].

Antioxidants are molecules that slow down or inhibit the oxidation of other molecules. Collagen peptides are considered to act as an antioxidant. Low-molecular-weight oligopeptides are widely used due to its excellent biocompatibility, easy biodegradability, and low antigenicity. Oral ingestion of oligopeptides increases the level of collagen-derived peptides in the blood and improves skin properties such as elasticity, moisture and TEWL. Daily intake of hydroylsed collagen decreases the skin aging process [38].

#### 3.4.3. Evaluation of Skin Anisotropy

In Table 3, the results of skin anisotropy index at zero, four and eight weeks of application of the cosmetic gel formulation and the values of the change in skin anisotropy index in each week are expressed in %. The table shows that the application of a cosmetic gel formulation with collagen hydrolysate has a positive effect on the resulting skin anisotropy index, when on the right side there was a decrease in the fourth week by 3.1 a.u. and then by 3.7 a.u. in the eighth week (as compared to the fourth week of application) and on the left side there was a decrease in the fourth week by 7.1 a.u. and then by 3.3 a.u. in the 8th week of application (as compared to 4th week of application). Overall, skin anisotropy decreased by 6.8 a.u. on the right side and by 10.4 a.u. on the left side. In numerical terms, there was a decrease from 23.9 ± 0.9 to 20.8 ± 0.6 a.u. (week 4) and then to 17.1 ± 0.6 a.u. (week 8) on the right side; and from 25.7 ± 0.5 to 18.6 ± 0.7 a.u. (week 4) and then to 15.3 ± 0.8 a.u. (week 8) on the left side.

A study by Tadini et al., in which the effect of a regularly applied cosmetic product with retinoic acid on anisotropy was tested, showed that already at week 4, then also at week 8, positive changes in skin anisotropy were observed, similar to the ones in this study. Thus, the mechanical properties of the skin were improved, and thus the signs of skin aging were reduced [42].

#### 3.4.4. Amount of Wrinkles

Table 4 shows the amount of wrinkles at zero, four and eight weeks of application of the cosmetic gel formulation with chicken collagen hydrolysate. We can see that with increasing application time, the amount of wrinkles decreases; there was an overall reduction of wrinkles by 21.8% and by 16.5% on the right and on the left side, respectively, of the monitored area. The values decreased from 15.6 ± 0.3 to 12.2 ± 0.1 (right eye area) and from 15.2 ± 0.3 to 12.7 ± 0.2 (left eye area). It follows that with regular application of chicken collagen hydrolysate twice a day (morning and evening), visible results can be expected in reducing the amount of wrinkles in the periorbital area.

Figure 9 and Figure 10 compared the visual change of skin condition in women aged 45 and 50 years. In a volunteer aged 45 years, there was an average decrease in wrinkles on both right and left temple from 12.2% to 10.6%. Similarly, a decrease from 17.7% to 16.6% was monitored in the volunteer aged 50; the numerical reduction was not so significant here, but the smoothing of the skin and the reduction of wrinkles is visible according to Figure 10. These two participants show very high initial values of the amount of wrinkles (compared to the average shown in Table 4), but the overall decrease in the amount of wrinkles in the periorbital area is visible, see Figure 9 and Figure 10.

## 4. Materials and Methods

The block diagram in Figure 11 describes the preparation of collagen hydrolysate, cosmetic gel formulation and its testing.

### 4.1. Appliances, Tools and Chemicals

The following appliances and materials were used: Meat cutter SPAR Mixer SP–100 AD–B (Gastrotip, Hradec Králové, Czech Republic), shaker LT2 (KavalierGlass, Praha, Czech Republic), electronic analytical laboratory balances Kern 770 (Verkon, Praha, Czech Republic), pH meter WTW 526 (Merck, Praha, Czech Republic), heating nest LTHS 250 (Fisher Scientific, Pardubice, Czech Republic), dryer WTB Binder E–28–TB1 (Binder GmbH, Tuttlingen, Germany), electric cooker with magnetic stirrer (Schott Instruments GmbH, Mainz, Germany), magnetic stirrer IKA Labortechnik PCT Basic with heating and with magnetic stirrer (IKA Labortechnik, Staufen im Breisgau, Germany), Erlenmeyer flask 2 L, PET bottles with a screw cap with a volume of 2 L, measuring cylinder 1000 mL, pipettes, syringes with distilled water, laboratory spoons and sticks, friction bowl, beakers, balloon, funnels, ordinary laboratory glassware, desiccator (Verkon, Praha, Czech Republic), MPA 10 station with individual probes, Tewameter TM 300^®^, Corneometer CM 825^®^, Skin–Visiometer SV 700^®^, Visioscope PC 35^®^, Revisometer RV 600^®^ (Courage & Khazaka GmbH, Köln, Germany), vacuum pump, laboratory stirrer Heidolph (VWR, Stříbrná Skalice, Czech Republic), mixer RZR 2020 (IKA Labortechnik, Staufen im Breisgau, Germany), paper double-sided adhesive rings, plastic cover foil, plastic cups and sticks, paper frames (SynCare, Brno, Czech Republic). Materials: pulp, cosmetic handkerchiefs, tampons (Česká lékárna holding, Brno, Czech Republic). Stationery: scissors, self-closing PE bags, adhesive tape, non-stick drying pads, PA fabric (McPen, Praha, Czech Republic). Kitchen utensils: metal sieves, plastic colander, pots, baking sheets (Tescoma, Praha, Czech Republic).

Enzyme Protamex^®^ (Novozymes, Copenhagen, Denmark); the enzyme is a *Bacillus* protease complex that has been developed for the hydrolysis of proteins intended for the food industry; Protamex^®^ meets the requirements for high purity [43]. Chemicals: distilled water, 0.03 M and 0.06 M NaOH, 0.2 M HCl, acetone, chloroform, ethanol, petroleum ether (Verkon, Praha, Czech Republic), Jarisch’s solution, Bioderma Sensibio^®^ H_2_O make-up remover (Česká lékárna holding, Brno, Czech Republic); two-component silicone set for the preparation of skin replicas (SynCare, Brno, Czech Republic).

The carbopol gel, a colorless and transparent gel of polyacrylic acid utilised as a cooling gel in pharmacy, was mixed in Fagron, Co. (Olomouc, Czech Republic). It is a weakly acidic gel (pH equals 4.5). The composition of the gel, according to International Nomenclature of Cosmetic Ingredients (INCI), is as follows: *carbomer, aqua, methylparaben, propylparaben* and *sodium hydroxide* [44].

### 4.2. Preparation of Collagen Hydrolysate and Cosmetic Gel Formulation

The composition of the raw chicken stomachs was the following: Dry matter content: 19.10 ± 0.05%; in dry matter: protein content 75.6 ± 0.8% (collagen content 82.8 ± 0.7%), fat content 21.70 ± 0.01%, minerals 3.900 ± 0.005%. Chicken stomachs were ground and homogenized into 3 mm particles, then washed thoroughly in water. The preparation of the collagen hydrolysate was performed according to optimized conditions previously published in the study by the authors [33]. The stomachs were treated in 0.2 M NaCl and 0.03 M NaOH for 2 h and 24 h, respectively, and then dried at 36.0 ± 0.2 °C for 34 to 36 h. They were further thoroughly degreased with petroleum ether and ethanol (1:1) in a ratio of 1:9 (100 g stomachs and 900 mL solvent mixture) for 48 h. Chicken collagen hydrolysate was extracted from the dried and defatted tissue by neutral treatment of the raw material with Protamex^®^. In the first phase, the chicken tissue was mixed with water in a ratio of 1:10 (100 g stomachs and 1000 mL water) and after 20 min the pH was adjusted to 6.3 ± 0.2 by adding 0.2 M HCl and using a solution of 0.03 or 0.06 NaOH. Subsequently, as the pH stabilized, 0.6% of Protamex^®^ enzyme was added, and the mixture was shaken intensively for 30 h at room temperature. In the second phase, the mixture was mixed with water in a ratio of 1:8 (100 g of stomachs and 800 mL of distilled water) and the tissue was heated to 66 °C (dt/dτ = 15 °C/min) and extracted at this temperature for 2 h. In the last step, the mixture was filtered several times through PA fabric and the resulting hydrolysate solution was dried in a thin film at 45.0 ± 0.2 °C for 24 to 36 h. All dried hydrolysate films were ground in a mortar and homogenized to a very fine powder of size particles 0.1–0.2 mm. To 1000 g of carbopol gel, chicken collagen hydrolysate with a concentration of 1.0% (*w*/*w*), i.e., 10 g of hydrolysate, was blended. The whole mixture was then homogenized on a stirrer at 2000 rpm for 10 min in a 1.5 L beaker.

The 1.0% (*w*/*w*) addition of collagen hydrolysate was chosen both on the basis of our own pilot experiments and according to the recommendations from the material sheets referring to the use of collagen hydrolysates from the database of cosmetic ingredients of SpecialChem [45,46]. From the results of our pilot studies on the application of 0.5% and 2.0% (*w*/*w*) additions of collagen hydrolysate to cosmetic creams and gels it was found that at both tested cosmetic formulations enriched with 0.5% (*w*/*w*) addition of collagen hydrolysate the higher value of skin hydration and lower value of TEWL were recorded. Compared to the cosmetic base matrixes, at formulations with 2.0% (*w*/*w*) addition of collagen hydrolysate no positive effect was observed for both monitored skin properties. These findings are in accordance with recommendations for the dosage of hydrolyzed collagens in cosmetic skincare formulations; the additions of 0.5–1.0% (*w*/*w*) of hydrolysates for skin preparations is reported to be optimal. The 1.0% (*w*/*w*) addition of collagen hydrolysate also complies with the safety recommendations for the use of proteins and peptides prepared from animal tissues in cosmetics [47].

### 4.3. Selection of Volunteers and Application of Cosmetic Gel Formulation

The experimental measurement involved 10 volunteers aged 50 ± 9 years. The measurements took place in February, March and April 2020, always in the same laboratory at a temperature of 21.0 ± 2.0 °C and a relative humidity of 38.0 ± 2.0%. The measurement took place in the periorbital area, i.e., on the right and left temple. The volunteers were acquainted with the course of the experiment, then filled in a questionnaire about their health condition and signed an informed consent. All volunteers agreed to be included in the study and had the right to withdraw at any time during the study without giving a reason. Before the experiment itself, but also during the experiment, the volunteers were asked not to apply any other cosmetic formulations to the skin in the periorbital area and not to undergo any cosmetic or aesthetic procedure. The selection of volunteers and the testing procedure were in accordance with international ethical principles of biomedical research with human participants [48]. The first measurement, or 0th measurement, was performed before using the cosmetic gel formulation under constant laboratory conditions (temperature, humidity) and the volunteers were left to rest for 15 min to acclimatize before starting the measurement. Before the measurement itself, the skin was first cleansed and degreased from impurities using Jarisch’s solution and Bioderma Sensibio^®^ H_2_O make-up remover. The second and third measurements took place after the 4th and 8th week of regular twice-a-day application in the same laboratory under the same conditions.

Study on the influence of pure carbopol gel on skin properties was not performed as in previous studies no significant effects on skin hydration and TEWL reduction were reported [49].

### 4.4. Diagnosing the Condition of the Skin after the Cosmetic Gel Application

#### 4.4.1. Skin Hydration

Skin hydration was determined using a corneometric probe (Courage & Khazaka GmbH, Köln, Germany), where 5 values were measured at each cleaned site (right and left temple), the calculation was performed at 3 values (the lowest and highest value was neglected). The CM 825^®^ corneometric probe is used to measure skin hydration, where the probe is placed on the periorbital area with gentle pressure; it is a very fast and easy measurement. According to the values, the scale of the corneometer records extremely dry (<30 c.u.), dry (30–45 c.u.) and normal (> 45 c.u.) skin; c.u. = corneometric units [50].

#### 4.4.2. Transepidermal Water Loss

Transepidermal water loss (TEWL) was determined using a tewametric probe, which measured 15 values at each irritated site (again, right and left temple), of which the first 5 values were neglected due to the effect of initial differences between skin and probe parameters (temperature, humidity). The TM 300^®^ tewametric probe is used to measure the amount of water evaporated from the skin and is related to Fick’s law, diffusion rate of mass transfer [51]. This is again a very quick and easy measurement to determine the skin’s natural barrier. The condition of the skin barrier function corresponds to the following TEWL scale: very good (0–10 g/m^2^/h), good (10–15 g/m^2^/h), normal (15–25 g/m^2^/h), tight (25–30 g/m^2^/h) and critical (>30 g/m^2^/h). The lower the TEWL, that healthier the skin; it can be concluded that TEWL values 0–10 g/m^2^/h indicate very healthy skin and values 10–15 g/m^2^/h point to healthy skin [52].

#### 4.4.3. Skin Elasticity

Using the RV 600^®^ revisometer, the elasticity of the skin can be measured; the device evaluates the speed of propagation of the acoustic wave through the skin surface. The measurement takes place at different angles and the results depend on both isotropy and anisotropy of the skin, which is affected by skin aging. The precondition is an increase in the elasticity of the skin (using a cosmetic gel formulation), i.e., a decrease in the RRT values (expressed as a.u. = arbitrary units), and thus an increase in the firmness of the skin. At the same time, the skin is unified, i.e., the anisotropy is reduced, because with increasing age, the anisotropy of mechanical properties increases in different directions of the skin. The measuring principle is based on the gentle pressure of two sensors on the skin, where one sensor emits a shock wave and the other sensor receives it. The RV 600^®^ revisometric probe was used to measure under the angle of 30° at the right and left temples, and the arithmetic mean and standard deviation were determined for each angle of rotation [53]. RV 600^®^ reviscometer has been recommended by previous studies as a sensitive and suitable tool for measuring the bio-mechanical properties of the skin [54,55].

#### 4.4.4. Skin Relief

To monitor changes in skin relief, a method to measure the profile (topography) of skin surface impressions was used. The system Skin–Visiometr SV 700^®^ was used for measurement; it works on the principle of measuring the intensity of light radiation (light transmission) transmitted by a silicone replica of the skin. The measuring principle is based on the passage of light through a very thin translucent special blue silicone replica. Silicone contains a constant concentration of blue pigment capable of absorbing radiation and the result is 2D (shades of gray) or 3D (color) images of skin relief and depth of wrinkles. The measurement is performed by inserting a replica between the light source and the CCD camera placed in parallel with it, which captures the transmitted light spectrum. The light is then absorbed depending on the thickness of the replica, depending on the light intensity and the length of the optical path of the light beam, thus, on the depth of wrinkles. Amount of light absorbed (*ϕ*_ex_) is determined by Lambert–Beer’s law:*ϕ*_ex_ = *ϕ*_in_ × e^(−kd)^(1)
where *ϕ*_in_ is the amount of incoming light, e is Euler’s number, k is the absorption constant and d is the length of the optical path.

The CCD camera (Courage & Khazaka GmbH, Köln, Germany), 2560 × 1920 Pixel resolution (5 MPixel) with measuring area of 9.0 × 6.7 mm (3.5 µm/pixel technical resolution), measures the light passing through the replica in gray levels and the acquired image is displayed in individual X, Y and Z coordinates, where X and Y describe the height and Z then the depth, which can be determined in the range from 20 to 300 µm.

Figure 12 shows all the steps, (a), (b) and (c), for scanning the replica. In the first phase, it was necessary to prepare a double-sided adhesive paper ring, a transparent cover foil, two-component silicone, a plastic cup, a vacuum pump, a plastic stick and a paper template. In the next step, the skin was cleansed of unwanted impurities and the skin area was defined for measuring skin relief (Figure 12a) and a double-sided adhesive paper ring was glued to this area (replica form, Figure 12b). In the second phase, the two silicone components, i.e., the base and the catalyst, were mixed in a ratio of 1:1 ratio for 20 s using a vacuum pump. Immediately after mixing, a small volume of silicone (1 to 2 drops) was transferred to the space where the replica form was located using a plastic stick. By applying a cover foil and pushing the excess silicone away from the eye, an equally thick silicone layer was obtained in all prepared replicas (Figure 12c). After 5 min, when the silicone hardened, the silicone mold was very carefully removed from the skin and glued to a special paper carrier, which was then inserted into the device Skin–Visiometr^®^ SV 700 so that the side covered by the cover plastic film lies in the direction of passage of the beam from the light source. In the last step, the data were processed and the skin roughness measurements R1, R2, R3, R4 and R5 were evaluated. These parameters are defined in DIN 4768 as Rt, Rmax, Rz, Rp and Ra, where parameter R1 (skin roughness, Rt) defines the distance between the highest peak and the lowest depression and thus represents the average of the distance between the highest peak and the lowest depression on the measured part of the skin relative to the reference length 1. The parameter R2 (maximum roughness, Rm or Rmax) is the largest local roughness of the roughness values of different segments. R3 (average roughness, Rz) is determined as the roughness arithmetic mean, calculated from five successive measurements of segments of the same length. R4 (roughness depth, Rp), smoothness depth (depression smoothness); this parameter describes the area above the reference profile closed by the line of the highest protrusion and divided by the center line determined by the average roughness. This area is then related to the length of the wrinkle. R5 (roughness arithmetic mean, Ra) is determined as the average deviation of the actual profile from the average profile; the area bounded by the reference profile and the center line determined by the average roughness [56]. All the parameters defined above are significantly dependent on the selected area of calculations, so to obtain comparable results, the same calculation area was set on the replicas taken before and after using the gel. The quantitative ratio between the parameters R1 (Rt) ≥ R2 (Rmax) ≥ R3 (Rz) ≥ R4 (Rp) ≥ R5 (Ra) was found both at 0 and subsequently at 4 and 8 weeks of application of the cosmetic gel formulation with collagen hydrolysate. Roughness parameters R1 to R5 were statistically evaluated using Microsoft Office Excel (Denver, CO, USA, 2010) when the arithmetic mean and standard deviation were determined for each of the parameters.

Skin–Visiometr SV 700^®^ can also calculate skin anisotropy from silicone replicas. Anisotropy is an important mechanical property of the skin and is caused by muscle contraction and the overall structure of the skin, which determines its elasticity. Anisotropy is another indicator of the aging process, where skin anisotropy increases with age [57]. Another in vivo biometric system for determining the viscoelastic properties of the skin and its anisotropy includes CutiScan CS 100 (Courage & Khazaka GmbH, Cologne, Germany), which is also used for this purpose by some authors [58].

#### 4.4.5. Amount of Wrinkles

The amount of wrinkles was determined using a PC 35 Visioscope^®^ with polarizing filters, where skin images were taken at 30 × magnification. This is a quick and easy analysis of the skin relief, an image is created and then an immediate analysis of wrinkles in the area in % can be performed.

### 4.5. Evaluation of Results and Statistical Analysis

The values of hydration, TEWL and skin elasticity were recorded and processed using the operating software of the MPA 10 station, CK–Multi Probe^®^ (Courage & Khazaka GmbH, Köln, Germany). Data obtained from silicone replicas were recorded and processed using the operating software of the MPA 10 station, Skin Visiometer^®^ SV 700 (Courage & Khazaka GmbH, Köln, Germany). The amount of wrinkles was recorded and processed using CSI–Complete Skin Investigation (Courage & Khazaka GmbH, Köln, Germany). The calculation of data and the creation of graphical outputs from the measurement of hydration properties, barrier functions, elasticity, roughness and relief of the skin was performed in Microsoft Office Excel (Denver, CO, USA, 2010). Arithmetic means and standard deviations were calculated for the values.

## 5. Conclusions

Collagen hydrolysate was prepared from poultry stomachs by a biotechnological process and was blended in an amount of 1.0% (*w*/*w*) to the carbopol gel to prepare cosmetic gel formulation. During eight weeks of regular topical application twice a day in volunteers in the periorbital area, the effect of cosmetic gel formulation on skin condition (skin hydration, transepidermal water loss, skin elasticity and change in skin relief), tested non-invasively, was monitored. First, an initial measurement was performed before the application of the cosmetic gel formulation, and then the gel was applied to cleansed skin in the area of the right and left temple. The scientific hypothesis was confirmed, as there was an overall increase in hydration of 11.82% (right side) and 9.45% (left side); there was also an overall decrease in TEWL of 25.70% and 17.80% on the right and the left side, respectively. A positive effect was also demonstrated on the elasticity of the skin because the values of resonance times decreased in individual directions, which indicates an increase in the elasticity of the skin (decrease in its toughness). Regarding the depth of the wrinkles (evaluated using Skin–Visiometr SV 700^®^) it was found that with increasing time of application of the cosmetic gel formulation with 1.0% chicken hydrolysate there is a reduction of wrinkles in the periorbital area by 35.4% (right temple) and by 41.2% (left temple). In general, it is important to keep in mind that the permeation capacity of substances depends on various factors, such as their physicochemical properties, time range of permeation, integrity, skin thickness, skin metabolism, site, time of application and amount of cosmetic formulation used. The biotechnological process of valorization of chicken stomachs into collagen hydrolysate represents an ecological and economical way to utilize food by-product as a sustainable source of secondary raw material with positive effects on the skin when added into cosmetic formulation. From the positive results of the bio-mechanical, hydrating and barrier properties of the skin after regular eight-week topical application of the gel formulation with the addition of chicken collagen hydrolysate it is obvious that chicken collagen hydrolysate is an effective cosmetic ingredient comparable to bovine and porcine collagen peptides and hydrolysates.

## Figures and Tables

**Figure 1 molecules-26-02021-f001:**
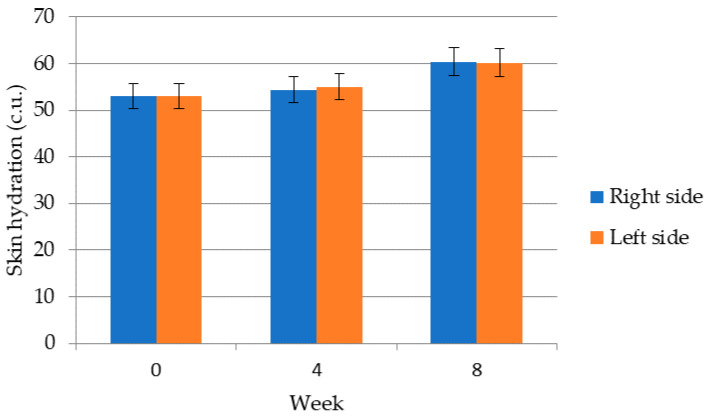
Skin hydration values of a group of volunteers in the periorbital area.

**Figure 2 molecules-26-02021-f002:**
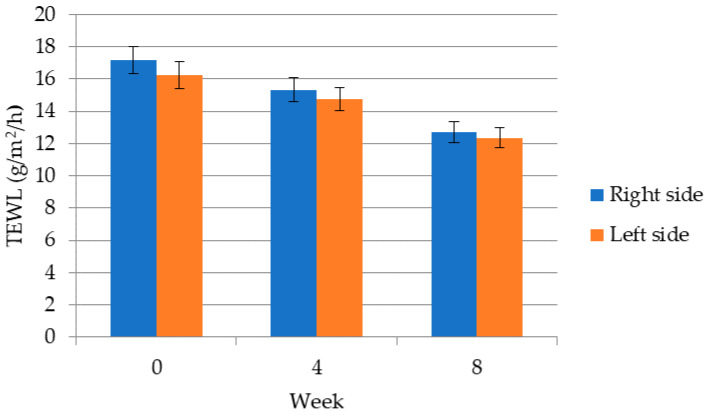
Transepidermal water loss (TEWL) values of a group of volunteers in the periorbital area.

**Figure 3 molecules-26-02021-f003:**
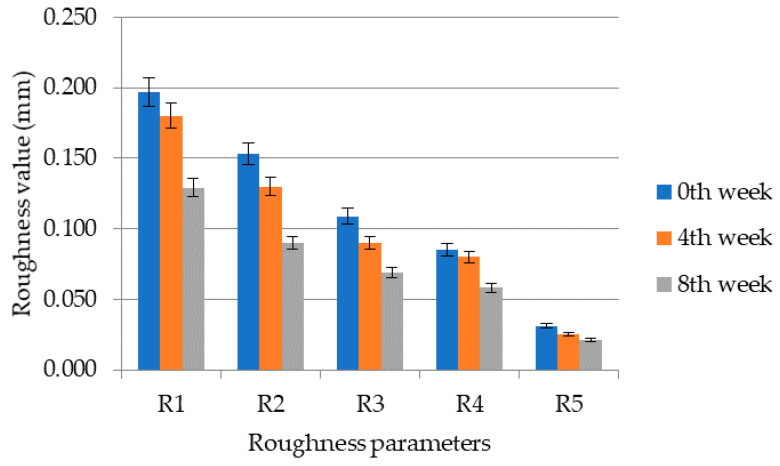
Values of roughness parameters R1 to R5 in individual weeks of measurement for the right temple area.

**Figure 4 molecules-26-02021-f004:**
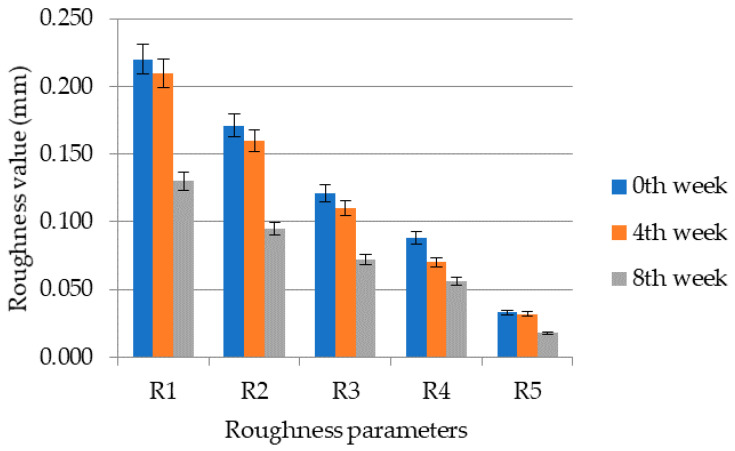
Values of roughness parameters R1 to R5 in individual weeks of measurement for the left temple area.

**Figure 5 molecules-26-02021-f005:**
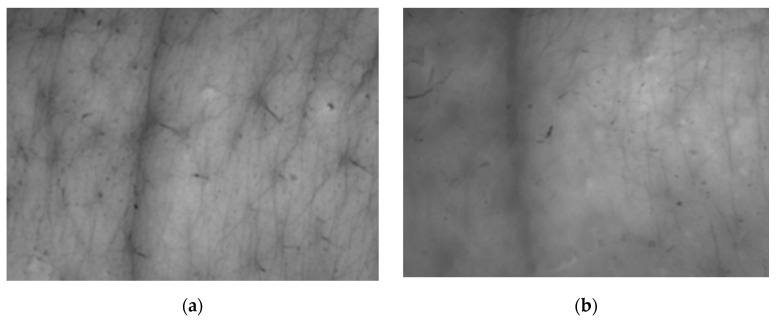
Comparison of 2D images of skin surface, woman aged 45 years: (**a**) Prior to applying the cosmetic gel formulation; (**b**) After eight weeks of application of the cosmetic gel formulation.

**Figure 6 molecules-26-02021-f006:**
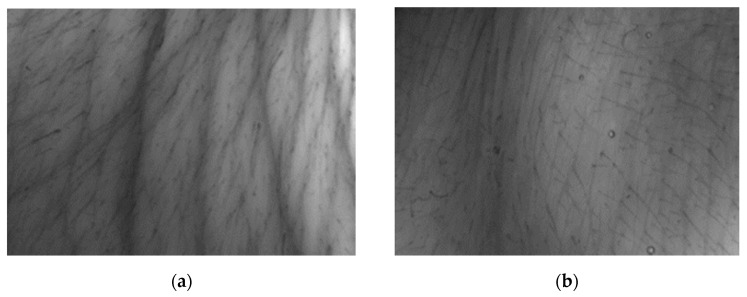
Comparison of 2D images of skin surface, woman aged 50 years: (**a**) Prior to applying the cosmetic gel formulation; (**b**) After eight weeks of application of the cosmetic gel formulation.

**Figure 7 molecules-26-02021-f007:**
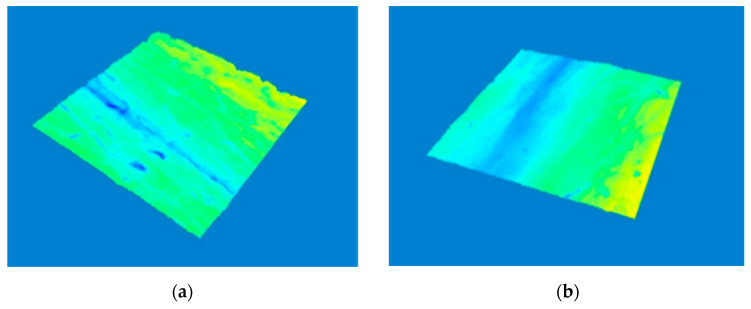
Comparison of 3D images of skin surface, woman aged 45 years: (**a**) Prior to applying the cosmetic gel formulation; (**b**) After eight weeks of application of the cosmetic gel formulation.

**Figure 8 molecules-26-02021-f008:**
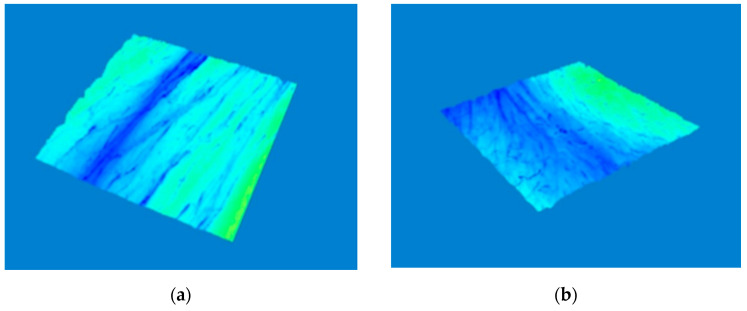
Comparison of 3D images of skin surface, woman aged 50 years: (**a**) Prior to applying the cosmetic gel formulation; (**b**) After eight weeks of application of the cosmetic gel formulation.

**Figure 9 molecules-26-02021-f009:**
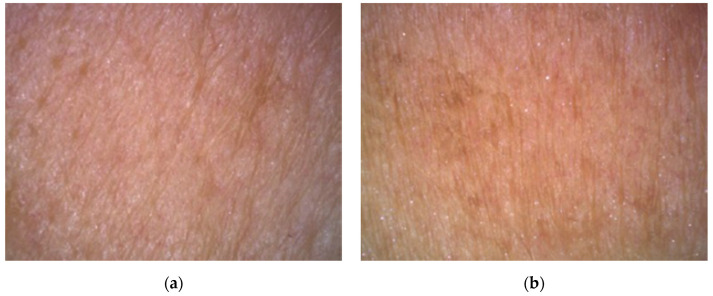
Skin roughness and wrinkle reduction (30 × magnification), woman 45 years: (**a**) Prior to applying the cosmetic gel formulation; (**b**) After eight weeks of application of the cosmetic gel formulation.

**Figure 10 molecules-26-02021-f010:**
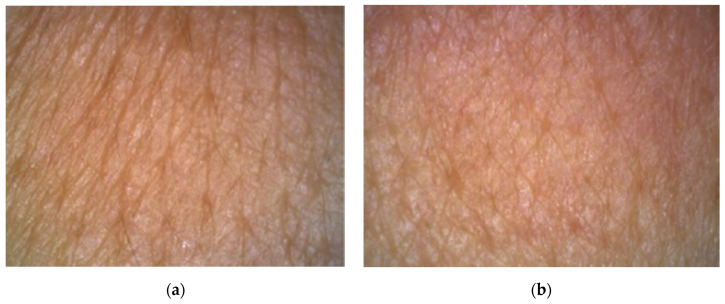
Skin roughness and wrinkle reduction (30 × magnification), woman aged 50 years: (**a**) Prior to applying the cosmetic gel formulation; (**b**) After eight weeks of application of the cosmetic gel formulation.

**Figure 11 molecules-26-02021-f011:**
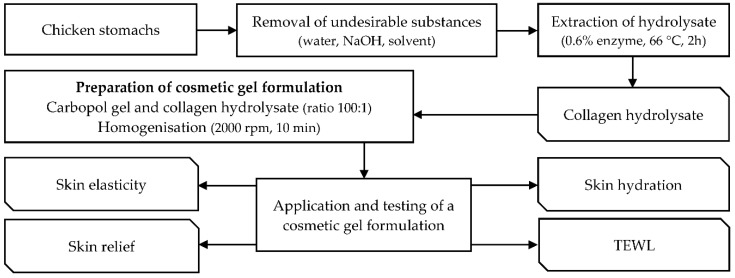
Block diagram showing the procedure of preparation of collagen hydrolysate, cosmetic gel formulation and its testing.

**Figure 12 molecules-26-02021-f012:**
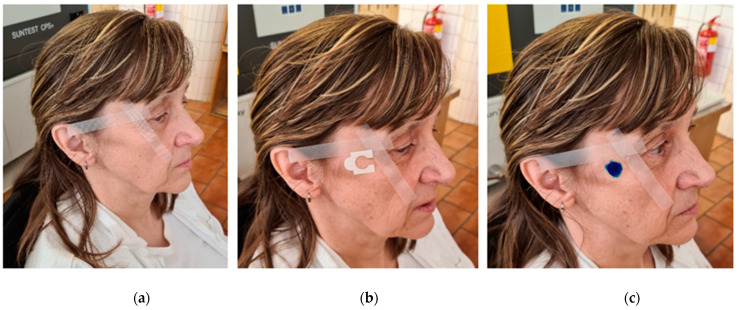
Silicone replica application procedure: (**a**) Defined area; (**b**) Glued paper ring; (**c**) Two-component silicone applied.

**Table 1 molecules-26-02021-t001:** Skin resonance time (RRT) values in the periorbital area in individual weeks of application for the right temple.

**Angle (°)**	**0**	**30**	**60**	**90**	**120**	**150**	**180**
**Week**	**RRT ± SD (a.u.)**						
0	235 ± 50	218 ± 50	216 ± 40	251 ± 60	325 ± 60	364 ± 50	296 ± 50
4	147 ± 20	285 ± 80	227 ± 40	162 ± 16	175 ± 30	192 ± 40	188 ± 30
8	171 ± 30	284 ± 90	167 ± 30	104 ± 60	264 ± 70	266 ± 70	238 ± 60
**Angle (°)**	**210**	**240**	**270**	**300**	**330**	**360**	
**Week**	**RRT ± SD (a.u.)**						
0	276 ± 40	260 ± 50	254 ± 60	238 ± 60	198 ± 30	213 ± 30	
4	187 ± 30	230 ± 70	206 ± 60	142 ± 15	158 ± 30	185 ± 40	
8	197 ± 60	236 ± 90	291 ± 90	200 ± 50	135 ± 16	176 ± 40	

**Table 2 molecules-26-02021-t002:** Skin RRT values in the periorbital area in individual weeks of application for the left temple.

**Angle (°)**	**0**	**30**	**60**	**90**	**120**	**150**	**180**
**Week**	**RRT ± SD (a.u.)**						
0	228 ± 50	234 ± 50	230 ± 30	262 ± 50	321 ± 60	344 ± 40	309 ± 50
4	156 ± 18	276 ± 80	242 ± 40	164 ± 15	183 ± 30	200 ± 40	197 ± 30
8	167 ± 20	288 ± 90	173 ± 30	200 ± 60	266 ± 70	265 ± 70	245 ± 60
**Angle (°)**	**210**	**240**	**270**	**300**	**330**	**360**	
**Week**	**RRT ± SD (a.u.)**						
0	262 ± 40	267 ± 50	245 ± 60	231 ± 60	211 ± 30	213 ± 30	
4	185 ± 19	234 ± 70	208 ± 50	145 ± 16	165 ± 30	186 ± 30	
8	195 ± 60	240 ± 80	301 ± 90	196 ± 50	137 ± 16	179 ± 40	

**Table 3 molecules-26-02021-t003:** Anisotropy index of the skin in the periorbital area in individual weeks of application.

Week	Right Side		Left Side	
	Skin Anisotropy Index ± SD (a.u.)	Change in Skin Anisotropy Index (a.u.)	Skin Anisotropy Index ± SD (a.u.)	Change in Skin Anisotropy Index (a.u.)
0	23.9 ± 0.9	---^1^	25.7 ± 0.5	---^1^
4	20.8 ± 0.6	–3.1	18.6 ± 0.7	–7.1
8	17.1 ± 0.6	–3.7	15.3 ± 0.8	–3.3

---^1^ Initial skin condition in the periorbital area, without application of cosmetic gel formulation with 1.0% chicken collagen hydrolysate.

**Table 4 molecules-26-02021-t004:** The amount of wrinkles in the periorbital area in individual weeks of application and the overall reduction of wrinkles.

Week	Right Side		Left Side	
	Amount of Wrinkles ± SD (%)	Overall Reduction of Wrinkles (%) ---^2^	Amount of Wrinkles ± SD (%)	Overall Reduction of Wrinkles (%) ---^2^
0	15.6 ± 0.3	---^1^	15.2 ± 0.3	---^1^
4	14.4 ± 0.6	7.7	14.5 ± 0.5	4.6
8	12.2 ± 0.1	21.8	12.7 ± 0.2	16.5

---^1^ Initial skin condition in the periorbital area, without application of cosmetic gel formulation with 1.0% chicken collagen hydrolysate. ---^2^ In comparison with the number of wrinkles at week 0.

## Data Availability

The data presented in this study are available on request from the corresponding author.

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
