# Peer review of "Collagen Hydrolysate Prepared from Chicken By-Product as a Functional Polymer in Cosmetic Formulation"

_molecules, 2021, doi:10.3390/molecules26072021_

Round 1

Reviewer 1 Report

The article "Collagen hydrolysate prepared from chicken by-product as a functional polymer in cosmetic formulation" by Prokopova et al describes various skin benefit efficacy parameters when formulated into a carbopol containing skin gel.  Although the new source of the collagen hydrosylate is interesting, the study is lacking in terms of providing any sort of comparitive controls with other collagen hydrosylate (from other sources) or with the carbopol control.  Only one formulation at 1% hydrosylate was tested without proper control formulations.  It is really hard to deduce the significance of the study as it currently stands. I would recommend the authors carry out additional studies including proper controls and additionally elucidate mechanistic understanding of any performance differences with the other types of hydrosylates. This journal does require in-depth analysis on a molecular mechanistic basis.  As it stands it might be more appropriate for an applied cosmetics journal.

Author Response

Dear Sir / Madam,

Thank you very much for revision of our paper and recommendations hoe to improve it. We did our best to revise our paper according to your suggestions. Changes in the manuscript are highlighted in red color. Below, there are responses to your comments.

Reviewer Comment: The article "Collagen hydrolysate prepared from chicken by-product as a functional polymer in cosmetic formulation" by Prokopova et al describes various skin benefit efficacy parameters when formulated into a carbopol containing skin gel.  Although the new source of the collagen hydrosylate is interesting, the study is lacking in terms of providing any sort of comparitive controls with other collagen hydrosylate (from other sources) or with the carbopol control.  Only one formulation at 1% hydrosylate was tested without proper control formulations.  It is really hard to deduce the significance of the study as it currently stands. I would recommend the authors carry out additional studies including proper controls and additionally elucidate mechanistic understanding of any performance differences with the other types of hydrolysates. This journal does require in-depth analysis on a molecular mechanistic basis.  As it stands it might be more appropriate for an applied cosmetics journal.

Authors Responses:

  1. The study did not compare the results of the observed skin properties after topical application of the gel formulation with 1.0% addition of collagen hydrolysate with carbopol control alone. This fact is newly explained in chapter 4.3. in the last paragraph.

"Study on the influence of pure carbopol gel on skin properties was not performed as in previous studies no significant effects on skin hydration and TEWL reduction were reported [49].“

From the literature study in the introductory part of the manuscript it is clear that the cosmetic carriers (base matrixes) without the addition of active cosmetic substances do not show such an effect on the bio-mechanical, barrier and hydrating properties of the skin. The aim of the study was to determine the condition of the skin after 8 weeks of application of carbopol gel with the addition of chicken collagen hydrolysate.

  1. A comparative comparison of the effect of a cosmetic formulation with chicken collagen hydrolysate with other types of collagen hydrolysates was not performed in this study as the positive effects (especially of porcine and bovine) of collagen in the form of collagen hydrolysates or oligopeptides have been confirmed in many studies. The aim of our study was to determine the hydrating, barrier and bio-mechanical properties of the skin after 8 weeks of application of carbopol gel with the addition of chicken collagen hydrolysate and to evaluate whether this type of hydrolysate can be an effective cosmetic ingredient and whether it can be used as a substitute for hydrolysates (peptides) produced from other collagen sources (bovine, porcine, fish). Our study proved it. In this sense, Introduction part has been extended (see the last paragraph) with other literature sources (9 new references) describing the effect of collagen hydrolysates on the skin not only after topical applications but also after oral administration. Furthermore, the results of our study were briefly contrasted in the section “5. Conclusion"- please see the last sentence.

  1. Regarding the testing of the cosmetic formulation with just 1.0% addition of collagen hydrolysate. This fact is newly explained in details in chapter 4.2 – please see the second paragraph; 3 literary references were newly added. According to the "Safety Assessment of Hydrolyzed Collagen", a concentration of hydrolyzed collagen of less than 5.0% is recommended for cosmetic products intended for application to the eye area; at higher doses, eye and eye area irritation may occur after topical application.

Reviewer 2 Report

The article is interesting. It is worth emphasizing that there is no information in the literature on the use of chicken hydrolysates in cosmetics. Undoubtedly, this is a new, unexplored area of ​​application.

Nevertheless, during reading, the following shortcomings were noted:

  • 3 - Point 3. Results and discussion should follow Chapter 4. Materials and methods. Please change the order. It will definitely improve understanding of the text by potential readers.
  • There is also no exact presentation of the gel recipe (eg in the table) taking into account the qualitative and quantitative analysis of all gel components. Although the diagram is described in Figure 11 (p. 11), it is in a very general way. Please write what specific carbopol was used (its INCI name, information, etc.). The contents of the gel should also be listed in the table. It would be advisable to write the information about obtaining the gel under the table.
  • Has the pH of the cosmetic gel been adjusted? If so, to what value? In the text L. 394. p.12 is given the value 4.5. Was it a sufficient pH value to get a strong gel for carbopol?
  • 442., P. 13, point 4.4 in my opinion should be called: Diagnosing the condition of the skin after the cosmetic gel application
  • 120 p.3 skin hydration - it is worth mentioning that above 45 c.u. the skin is sufficiently moisturized. 10 people participated in the category. Is the chart an average of 10 people? Were there people who initially had values lower ​​than 45 c.u.?
  • For TEWL values ​ between 0 and 10 points indicate very healthy skin, and between 10 and 15 points indicate healthy skin. Perhaps it is worth adding it in the text. Is the chart an average of 10 people?
  • 462 p. 13. 4.4.3. The elasticity of the skin. Increasing the elasticity of the skin and thus increasing (not reducing) skin firmness is caused by a decrease in the RRT value.
  • Conclusion - I do not like the expression that the hydrolyzate was added to the gel. It looks as if the hydrolyzate has been added to the commercial gel. I also do not fully agree with the wording supplement, but rather, in the case of cosmetics, an active substance.

Author Response

Dear Sir / Madam,

Thank you very much for revision of our paper and detailed recommendations how to improve it. We did our best to revise our paper according to your suggestions. Changes in the manuscript are highlighted in red color. Below, there are responses to your comments.

Reviewer Comment: Point 3. Results and discussion should follow Chapter 4. Materials and methods. Please change the order. It will definitely improve understanding of the text by potential readers.

Response: The manuscript was prepared according to the latest Molecules template, which prescribes the arrangement of individual chapters in the text. Chapter “3. Results and Discussion” is placed before chapter “4. Materials and Methods ”. That's why we followed these rules.

Reviewer Comment: There is also no exact presentation of the gel recipe (eg in the table) taking into account the qualitative and quantitative analysis of all gel components. Although the diagram is described in Figure 11 (p. 11), it is in a very general way. Please write what specific carbopol was used (its INCI name, information, etc.). The contents of the gel should also be listed in the table. It would be advisable to write the information about obtaining the gel under the table.

Response: Detailed information on the composition and preparation of carbopol gel is newly given in chapter 4.1. – please see the last paragraph.

Reviewer Comment: Has the pH of the cosmetic gel been adjusted? If so, to what value? In the text L. 394. p.12 is given the value 4.5. Was it a sufficient pH value to get a strong gel for carbopol?

Response: The carbopol gel was mixed and supplied by Fagron, Co. (Olomouc, Czech Republic). The pH of the gel (4.5) was guaranteed and was not adjusted.

Reviewer Comment: 442., P. 13, point 4.4 in my opinion should be called: Diagnosing the condition of the skin after the cosmetic gel application.

Response: Chapter title 4.4. has been changed as recommended.

Reviewer Comment: 120 p.3 skin hydration - it is worth mentioning that above 45 c.u. the skin is sufficiently moisturized. 10 people participated in the category. Is the chart an average of 10 people? Were there people who initially had values lower ​​than 45 c.u.?

Response: Information regarding the sufficient skin moisturization for hydration above 45 c.u. is newly provided – please, see the second sentence of chapter 3.1. The values given are the average data of all 10 volunteers. In two volunteers, lower skin hydration was found at the beginning of the experiment, at 43 and 41 c.u., which is slightly below 45 c.u. Their skin condition was therefore marked as dry. In further measurements in these two volunteers, skin hydration increased to 47 and 47%, respectively, at 60 c.u.

Reviewer Comment:  For TEWL values ​ between 0 and 10 points indicate very healthy skin, and between 10 and 15 points indicate healthy skin. Perhaps it is worth adding it in the text. Is the chart an average of 10 people?

Response: The values given for TEWL are the average data of all 10 volunteers. According to the scale from Courage & Khazaka Electronic GmbH: Tewameter® TM 300 the skin condition is as follows: very good (0–10 g/m2/h), good (10–15 g/m2/h), normal (15–25 g/m2/h), tight (25–30 g/m2/h) and critical (> 30 g/m2/h). Of course, for intact corneum, TEWL values are lower. In the revised manuscript it is included that the values between 0–10 g/m2/h represent very healthy skin and the values between 10–15 g/m2/h represent healthy skin.

Reviewer Comment: 462 p. 13. 4.4.3. The elasticity of the skin. Increasing the elasticity of the skin and thus increasing (not reducing) skin firmness is caused by a decrease in the RRT value.

Response: Thanks for the remark. The error in the text has been revised.

Reviewer Comment: Conclusion - I do not like the expression that the hydrolyzate was added to the gel. It looks as if the hydrolyzate has been added to the commercial gel. I also do not fully agree with the wording supplement, but rather, in the case of cosmetics, an active substance.

Response: The sentence concerning the addition of collagen hydrolysate to the carbopol gel was reformulated as recommended – please see the first sentence of the chapter “5. Conclusions”. We also avoided using the word "supplement", instead the recommended word collocations "active substance", "active ingredients" etc. are used in the text.

Reviewer 3 Report

The manuscript is written very clearly. Methods selected are suitable although standardd in the filed. Lab condistions and pre-treatment of skin areas have been taken into account. Results are discussed in comparison of ohter groups. Results are quite impressive, especially those on skin roughness. The only concer I have about this study is that teh study was not done in a vehicle-controlled way. Unfortunately, this is a rather major concern that cannot be solved after closing the study. Maybe the authhors can elaborate on this in a theoretical way. Which results would have been anticipated using the carbomer-gel alone? Without chicken hydrolysate?  What is know from the literature?

Author Response

Dear Sir / Madam,

Thank you very much for revision of our paper and suggestions hoe to improve it. We did our best to revise our paper according to your comments. Changes in the manuscript are highlighted in red color. Below, there are responses to your comments.

Reviewer Comment: The manuscript is written very clearly. Methods selected are suitable although standard in the filed. Lab conditions and pre-treatment of skin areas have been taken into account. Results are discussed in comparison of other groups. Results are quite impressive, especially those on skin roughness. The only concern I have about this study is that the study was not done in a vehicle-controlled way. Unfortunately, this is a rather major concern that cannot be solved after closing the study. Maybe the authors can elaborate on this in a theoretical way. Which results would have been anticipated using the carbomer-gel alone? Without chicken hydrolysate?  What is know from the literature?

Authors Responses:

  1. The study did not compare the results of the observed skin properties after topical application of the gel formulation with 1.0% addition of collagen hydrolysate with carbopol control alone. This fact is newly explained in chapter 4.3., please see the last paragraph. "

"Study on the influence of pure carbopol gel on skin properties was not performed as in previous studies no significant effects on skin hydration and TEWL reduction were reported [49].“

From the literature study in the introductory part of the manuscript it is clear that the cosmetic carriers (base matrixes) without the addition of active cosmetic substances do not show such an effect on the bio-mechanical, barrier and hydrating properties of the skin. The aim of the study was to determine the condition of the skin after 8 weeks of application of carbopol gel with the addition of chicken collagen hydrolysate. Furthermore, it is known from the literature that the amount of bioactive collagen in a cosmetic formulation compared to placebo has an effect on the reduction of wrinkles in the eye area. In a study by Proksch et al. there was a reduction in eye wrinkles by almost 18% compared with placebo. At the same time, it has also been found that with increasing amounts of collagen, skin properties such as elasticity, moisture, but also TEWL improve; this means that with increasing amounts of collagen the slower skin aging occurs. These facts are newly discussed in chapter 3.4.2.

  1. We have expanded the chapter “1. Introduction” by 9 new References focused on the effect of collagen hydrolysates in both topical applications and oral dosing on the skin – please, see the last paragraph.

  1. The discussion part of the manuscript has also been improved in terms of confrontations with new literature data.

Round 2

Reviewer 1 Report

The authors have adequately addressed by concerns

Reviewer 3 Report

All issues have been adressed. The paper may now be published as is.